# ANGPTL4: A Comprehensive Review of 25 Years of Research

**DOI:** 10.3390/cancers17142364

**Published:** 2025-07-16

**Authors:** Pedro Ramos, Qiongyu Shi, Jeremy Kleberg, Chandra K. Maharjan, Weizhou Zhang, Ryan Kolb

**Affiliations:** 1Department of Pathology, Immunology and Laboratory Medicine, College of Medicine, University of Florida, Gainesville, FL 32610, USA; pedroramos@ufl.edu (P.R.); qshi1@ufl.edu (Q.S.); jkleberg@ufl.edu (J.K.); cmaharjan@ufl.edu (C.K.M.); zhangw@ufl.edu (W.Z.); 2Department of Biochemistry and Molecular Biology, College of Liberal Arts and Sciences, University of Florida, Gainesville, FL 32610, USA; 3UF Health Cancer Center, University of Florida, Gainesville, FL 32610, USA

**Keywords:** angiopoietin-like 4, lipid metabolism, angiogenesis, vascular permeability, metastasis, biomarker, therapeutic target

## Abstract

Angiopoietin-like 4 is a secreted protein with multiple biological functions and has been shown to play a key role in several human diseases. Here we review the research into the function and role of ANGPTL4 in human diseases, with an emphasis on its role in cancer.

## 1. Structure and Function

*ANGPTL4* was first identified in adipose tissue and later found to be widely expressed in several tissues including the liver, skeletal muscle, small intestine, placenta, heart, and kidney [1,2,3,4,5,6,7,8]. *ANGPTL4* expression was first identified to be regulated by peroxisome proliferator-activator receptors (PPARs), PPARα and PPARγ [1,2,3]. *ANGPTL4* expression is increased during a fasting state—a process that is mediated via glucocorticoids and the glucocorticoid receptor [9,10]. Several other transcription factors have also been shown to regulate *ANGPTL4* expression, including hypoxia-inducible factor 1 (HIF1), transforming growth factor beta (TGF-β), and inflammatory cytokines such as interleukin (IL)-1β and tumor necrosis factor α (TNFα) [11,12,13,14].

*ANGPTL4* encodes a glycoprotein that consists of an N-terminal signal peptide required for secretion, an N-terminal coiled-coil domain, a C-terminal fibrinogen-like domain, and a linking region between the two domains [15,16]. The linking region contains a substrate recognition site that is cleaved by pro-protein convertases (PCs) releasing the N-terminal peptide containing the coiled-coil domain (nANGPTL4) and the C-terminal peptide with the fibrinogen-like domain (cANGPTL4) (Figure 1A) [17,18,19]. The full length ANGPTL4 forms an oligomer via disulfide bonds between conserved cysteines within the N-terminus region prior to proteolytic cleavage; thereafter cANGP4TL4 circulates as a monomer and nANGPTL4 maintains the oligomeric structure [20]. Interestingly, the processing of ANGPTL4 varies depending on the tissue of expression; the cleaved nANGPTL4 is secreted in the liver, while adipose tissue can secrete the full length protein [19], indicating that the processing of ANGPTL4 may be important for its tissue specific function. Furin, PC5/6, PACE4, and PC7 are some of the PCs that can regulate ANGPTL4 processing. The tight regulation of these enzymes within tissues, development, hypoxia, and disease may contribute to ANGPTL4-specific functions [18,21].

ANGPTL4 belongs to the ANGPTL family of proteins, which share some structural and functional similarities, but have some uniqueness in their expression pattern and how they are regulated in performing their biological functions [22]. There are eight proteins in the ANGPTL family of proteins, with ANGPTL1-7 being distinguished structurally by an N-terminal coiled-coil domain and a C-terminal fibrinogen-like domain, and with ANGPTL8 being an atypical member having neither domain [22,23]. While structurally similar to angiopoietin proteins (ANG1 and 2), ANGPTL proteins do not interact with the receptors for ANG1 and 2 (tyrosine kinase with immunoglobulin-like and EGF-like domains 1 (Tie1) and Tie2), and therefore are considered orphan ligands [2,24,25]. ANGPTL proteins exhibit multiple functions and are involved in lipid metabolism, angiogenesis, inflammation, and hematopoietic stem cell activity [22]. As discussed in more detail in Section 1.1, ANGPTL3, 4, and 8 have been extensively studied in the regulation of lipid metabolism and coordination in regulating lipoprotein lipase activity during feasting and fasting states [4]. ANGPTl1 is primarily expressed in the liver and muscle and is a potent inhibitor of angiogenesis [26]. ANGPTL2 is expressed in several tissues, including adipose tissue and skeletal muscles, and secreted into circulation where it has been shown to promote angiogenesis and endothelial cell migration [22]. Circulating levels of ANGPTL2 have been associated with inflammation, obesity, and insulin resistance [22,27]. ANGPTL5 has no mouse ortholog, and little is known about its biological function, although it has been reported to be expressed in human cord blood hematopoietic stem cells [28,29]. Little is known about ANGPTL6 function, but it has been shown to be upregulated in patients with metabolic syndrome [30]. ANGPTL7 is expressed in the eyes and neuronal tissue, and has been reported to play a role in regulating intraocular pressure and the pathology of glaucoma [31].

### 1.1. Regulation of Lipid Metabolism and Triglyceride Uptake

ANGPTL4 is well-known for its role in lipid metabolism and triglyceride uptake. The E40K variant of ANGPTL4 is the most common loss-of-function mutation in humans, resulting in an unstable protein and leading to higher levels of plasma high density lipoprotein (HDL)-C and lower plasma triglycerides (TGs) [32,33,34]. Early studies found that ANGPTL4 regulates TG levels by inhibiting lipoprotein lipase (LPL) via the coiled-coil domain in nANGPTL4 [3,35,36]. LPL, which metabolizes TGs in circulating lipoproteins into cholesterol and fatty acids, is a key contributor to TG partitioning, which allows for the rapid utilization or storage of lipids as needed [37,38]. Several mechanisms for how ANGPTL4 inhibits LPL have been proposed. ANGPTL4 can mediate the unfolding of LPL, leading to its cleavage by proprotein convertases and subsequent degradation. ANGPTL4 can also decrease the affinity of LPL toward glycosylphosphatidylinositol-anchored high density lipoprotein-binding protein 1 (GPIHBP1), which is critical for the recruitment and anchoring of LPL to the surface of endothelial cells and facilitating LPL metabolism of TGs in circulating lipoproteins [15,17,38,39,40,41,42].

ANGPTL4 cooperates with ANGPTL3 and ANGPTL8 to regulate LPL, thus playing a critical role in the partitioning of TG to allow rapid utilization or storage of lipids during feasting and fasting states [4,38] (Figure 1B). ANGPTL3, which can inhibit LPL and endothelial lipase activity, is expressed in the liver, while ANGPTL8 is expressed in the liver and adipose tissue [43,44]. In the liver, ANGPTL8 forms a complex with ANGPTL3, which activates the LPL inhibitor function of ANGPTL3 and is secreted into the circulation [45,46,47,48,49,50,51,52,53,54]. In adipose tissue, ANGPTL8 forms a complex with ANGPTL4, reducing its ability to inhibit LPL [53,55]. When in a fasting state, ANGPTL4 expression is quickly upregulated by glucocorticoids in white adipose tissue, while the levels of ANGPTL8 are rapidly reduced [3,17,47,49,50,56,57,58]. In contrast, ANGPTL3 levels remain relatively constant [53] (Figure 1B). Thus, LPL activity is inhibited in adipose tissue and the content of the circulating ANGPTL3/8 complex is reduced, leading to increased circulating lipoproteins and increased LPL activity in oxidative tissue that allows them to utilize TGs. In a fed state, ANGPTL8 levels are increased, which leads to increased levels of the circulating ANGPTL3/8 complex and ANGPTL4/8 complex in adipose tissue. Thus, in a fed state, lipoproteins are funneled to adipose tissue where TGs are stored [38,59]. Additionally, ANGPTL4 can promote lipolysis in adipocytes, resulting in the release of fatty acids, though this appears to be a function of cANGPTL4, and the mechanism is not entirely known [43,44,60].

Apart from its LPL-dependent role in regulating lipid metabolism at a systemic level during feeding and fasting periods, recent studies have indicated that ANGPTL4 may act as a metabolic switch at a local cellular or tissue level. Using an endothelial cell conditional knockout of ANGPTL4, Chaube et al. showed that loss of ANGPTL4 in endothelial cells results in decreased angiogenesis and increased lipid metabolism [61]. Further studies indicated that ANGPTL4 can inhibit endothelial lipase (EL), increasing the bioavailability of free fatty acids; thus, loss of ANGPTL4 increased fatty acid uptake and oxidation, a metabolic state that favors endothelial cell quiescence. Concordantly, when ANGPTL4 is present, EL is inhibited, decreasing the bioavailability of free fatty acids and resulting in decreased oxidation and increased glycolysis, a metabolic state that favors angiogenesis [61]. Interestingly, we found similar results in clear cell renal cell carcinoma (ccRCC) cells. ANGPTL4 is significantly elevated in ccRCC. CRISPR Cas9-mediated knockout of ANGPTL4 in ccRCC cells resulted in an enrichment of genes related to lipid and cholesterol metabolism, fatty acid uptake, and beta-oxidation, indicating a potential increase in lipid metabolism in ccRCC cells lacking ANGPTL4 [62]. Loss of ANGPTL4 in ccRCC cells increases lysosomal acid lipase (LAL) activity, suggesting a similar mechanism as seen in endothelial cells but via inhibition of LAL instead of EL [61,62]. These data indicate that ANGPTL4 may play a role in metabolic reprogramming at a cellular or tissue level by modulating the bioavailability of free fatty acids and cholesterol via regulating the activity of various lipases.

### 1.2. Angiogenesis and Vascular Permeability

As mentioned previously, ANGPTL4 has multifaceted functions. While the regulation of lipid metabolism via inhibition of lipase activity is a function of nANGPTL4, its other known functions are generally mediated by the C-terminus fibrinogen-like domain (cANGPTL4) (Figure 1C). Due to its structural similarity to angiopoietin proteins, which have an integral and well-defined role in vascular biology, one of the most well-studied roles of ANGPTL4 is its role in vascular biology. The role of ANGPTL4 in vascular biology is complex and context-dependent, with both pro- and anti-angiogenic functions having been described [63,64]. Ito et al. first described that ANGPTL4 could inhibit endothelial cell proliferation, chemotaxis, and tube formation [24]. Later, Cazes et al. showed that ANGPTL4 accumulates in the subendothelial extracellular matrix, where it is either cleaved into nANGPTL4 and cANGPTL4 and remains soluble, or the full-length protein interacts with the extracellular matrix and becomes immobilized [65]. They further showed that immobilized ANGPTL4 inhibited endothelial cell adhesion, migration, sprouting, and tube formation [65] (Figure 1D). In corroboration of this study, a later study from the same laboratory by Chomel et al. showed that the N-terminus coiled-coil domain could interact with the extracellular matrix while protecting ANGPTL4 from proteolytic cleavage and facilitating its anti-angiogenic function [66]. Other studies have shown that cANGPTL4 can inhibit angiogenesis via the RAF/MEK/ERK signaling pathway [67,68]. In contrast, numerous studies have found a pro-angiogenic function for ANGPTL4, and more specifically, cANGPTL4 (Figure 1C). In 2000, Kim et al. showed that ANGPTL4 protects endothelial cells from apoptosis [2]. Other studies have found that ANGPTL4 can promote angiogenesis in various pathological conditions, including arthritis, renal cell carcinoma, and breast cancer [69,70,71]. Kolb et al. found that adipocyte-derived ANGPTL4 could promote angiogenesis in models of obesity-driven breast cancer and that antibodies against cANGPTL4 can inhibit this process [13]. Sodhi et al. demonstrated that recombinant human ANGPTL4 promotes in vitro tube formation of immortalized human microvascular endothelial cells and neovascularization in vivo [72]. Another study found that ANGPTL4 can promote angiogenesis during wound healing via keratinocyte to endothelial cell crosstalk mediated by inducible nitric oxide synthase (iNOS)-produced nitric oxide [73]. While the fibrinogen-like domain of ANGPTL4 is similar to that of the corresponding domain in angiopoietin 1 and 2, it is shorter and does not interact with the receptors for angiopoietin 1 and 2, Tie 1 and 2, which mediate their pro-angiogenic signaling. Thus, the pro-angiogenic function of cANGPTL4 is through a different mechanism [2,24,25]. Indeed, cANGPTL4 remains an orphan ligand, and the mechanism(s) by which ANGPTL4 promotes angiogenesis under what conditions remain to be fully elucidated.

Apart from its role in the regulation of angiogenesis, strong evidence supports the hypothesis that cANGPTL4 can promote vascular permeability [74]. The Massague laboratory first showed that ANGPTL4 could promote lung metastasis by disrupting the integrity of the cell–cell junctions of lung endothelial cells, facilitating the trans-endothelial migration of cancer cells [12]. Huang et al. later demonstrated that cANGPTL4 modulated endothelial junction integrity via interactions with VE-cadherin and claudin-5 and integrin-mediated signaling [75] (Figure 1C). They found that cANGPTL4 activates Rac1/PAK signaling via integrin α5/β1, leading to weakened cell–cell adhesion and increased vessel permeability. This allows cANGPTL4 to more readily interact with VE-cadherin and claudin-5, resulting in their declustering and internalization, increased nuclear β-catenin, and further disruption of the vascular barrier. While most studies have found that ANGPTL4 promotes vessel permeability, a few studies have demonstrated a protective role for ANGPTL4 in vascular integrity. ANGPTL4 was found to inhibit vessel permeability and prevent metastasis [24,76]. Furthermore, ANGPTL4 has been shown to protect vascular integrity in models of ischemic stroke [77,78]. While ANGPTL1–7 all have a C-terminal fibrinogen-like domain, their expression patterns vary widely and they have been reported to have different functions [22]. More studies are needed to clearly define the role of ANGPTL4 and other ANGPTL proteins in modulating vascular cell–cell interactions and their effects in various pathologies.

### 1.3. Other Functions

Apart from its roles in regulating lipid metabolism and in vascular biology, several other functions for ANGPTL4 have been reported, including those related to stem cell expansion and inflammation. ANGPTL4, along with the other members of the ANGPTL family of proteins (1, 2, 3, 6 and 7), have been reported to increase in vitro expansion of hematopoietic stem cells (HSCs) in the presence of growth factors [79,80]. In the absence of growth factors, ANGPTLs promote survival of HSCs but not their proliferation [79]. Schumacher et al. also reported that ANGPTL4 could expand the frequency of granulocyte-macrophage progenitor cells in bone marrow and facilitate the differentiation of megakaryocytes from HSCs [81]. Another study found that ANGPTL4 is upregulated at the hypoxic site of a bone fracture and could promote the osteoblastic differentiation of a preosteoblast cell line [82]. ANGPTL4 has also been reported to play a role in the regulation of inflammation, both positively and negatively depending on the tissue and the condition. ANGPTL4 is upregulated in mesenchymal stem cells under inflammatory conditions, where it suppresses the expansion of pro-inflammatory macrophages [14]. ANGPTL4 promotes M2 polarization of Kupffer cells and suppresses inflammation by inhibiting NF-κB in liver transplant models [83]. In LPS or influenza pneumonia induced lung injury, ANGPTL4 is upregulated in lung alveolar epithelial cells, where it promotes immune infiltration and inflammation. Loss of ANGPTL4 results in decreased lung inflammation and reduced lung injury [84,85]. These functions of ANGPTL4 in promoting survival or differentiation of various stem cell populations and in regulating inflammation need to be studied further to elucidate the mechanism behind these functions and how this affects various pathophysiological conditions.

## 2. ANGPTL4 in Human Diseases

### 2.1. Cardiovascular Diseases

Atherosclerosis is the buildup of plaques on the vessel wall, which contain a variety of biomolecules, including cholesterol and lipids, that narrow the arteries, resulting in less oxygen delivery and inflammation [86,87]. When this occurs within the heart, it is referred to as coronary artery disease (CAD) [86,87]. Within the United States, CAD accounts for one in four deaths, making it the leading cause of death in the country and third on the list worldwide [88]. Important risk factors of CAD include obesity, hypertension, and hyperlipidemia [88]. Smart-Halajko et al. initially showed a correlation between ANGPTL4 levels and risk for CAD; however, after adjusting for other CAD risk factors, the correlation was no longer significant [89]. Despite this initial study, other studies were conducted with a more representative sample population, and they confirmed a relationship between ANGPTL4 levels and the risk of CAD, disputing the findings of the first study [90,91,92]. Additional testing showed that lowering ANGPTL4 levels also led to decreased triglyceride and increased high-density lipoprotein (HDL) levels, both of which are beneficial for reducing CAD risk [90,92]. When investigating the role of genetic variants of ANGPTL4 on the risk of CAD, it was shown that loss of function variants correlated with decreased CAD risk [34]. Specifically, the E40K loss of function variant confirmed this association, particularly in white populations [93]. The rs2967605T variant, which is associated with higher HDL levels, also showed a decreased risk for CAD within the Southern Chinese Han population [94]. Collectively, these studies confirm that there is a significant relationship between ANGPTL4 levels and risk of CAD [92].

ANGPTL4 has been shown to decrease the formation of atherosclerotic plaques in mouse models [95]. The authors showed that ANGPTL4 increases collagen uptake in atherosclerotic plaques, leading to plaque stabilization and decreased disease progression [95]. Moreover, ANGPTL4 decreases Kruppel-like factor 4 expression in vascular smooth muscle cells (VSMCs), helping maintain a more contractile phenotype rather than the dedifferentiated synthetic phenotype found within atherosclerotic plaques [95]. Synthetic VSMCs are more proliferative and migratory. They secrete inflammatory cytokines, extracellular matrix components, and remodeling proteins [96,97]. The switching of VSMCs from a contractile phenotype to a synthetic one is known to promote the formation and progression of atherosclerosis and has been reviewed in several papers [97,98]. Additionally, ANGPTL4 was correlated with lower expression of inflammatory markers, consistent with decreased macrophage and monocyte levels within the plaques [95,99]. During the progression of atherosclerosis, macrophages infiltrate into the arterial wall and uptake excess lipids, leading to the formation of foam cells, which play a crucial role in the pathogenesis and progression of atherosclerosis [100,101,102]. ANGPTL4 suppresses foam cell formation by inhibiting LPL activity, downregulating CD36 expression and increasing cholesterol expulsion from macrophages [103,104]. By decreasing the inflammatory response, mitigating foam cell formation, increasing plaque stability, and increasing the contractile phenotype of VSMCs, ANGPTL4 expression can combat atherosclerotic plaque progression.

CAD and atherosclerosis can lead to myocardial infarction (MI) and ischemic events. Following an MI, patients may experience a complication called no-reflow, which indicates that the vessel blockage may still be present even after it has been cleared. This phenomenon has been associated with ANGPTL4 levels, where decreased levels indicate a higher likelihood of no-reflow [105]. ANGPTL4’s role in promoting angiogenesis may explain the connection, but the exact mechanism is unclear and needs further study. ANGPTL4 has also been associated with the regulation of inflammation and repair following MI and ischemia. Following an MI or ischemic event, MSCs can secrete ANGPTL4, which can suppress polarization of macrophages to an inflammatory state and promote an anti-inflammatory phenotype that aids in repair [14]. ANGPTL4 has been shown to promote vessel integrity, increasing angiogenesis and stabilization of endothelial cells following MI and ischemia [105,106]. Furthermore, when ANGPTL4 was delivered via a hydrogel patch, providing sustained levels within the heart, it enhanced vascularization and promoted the recruitment of M2 macrophages [107]. These studies suggest that ANGPTL4 can have a positive impact on MI and ischemia by suppressing inflammation and promoting repair. However, a study of patients recorded in the Chinese PLA Heart Failure Registry found a correlation between elevated levels of ANGPTL4 and increased risk of heart failure [108]. ANGPTL4 has also been shown to trigger cardiomyocyte apoptosis through the focal adhesion kinase (FAK) pathway, which could potentially cause additional heart damage [109]. The studies indicating that ANGPTL4 may be able to protect against complications of atherosclerosis and CAD have led some to hypothesize that ANGPTL4 may be useful as a potential therapeutic. However, its potential correlation with heart failure and cardiomyocyte apoptosis needs to be considered.

### 2.2. Retinopathies

Retinopathy encompasses a group of retinal disorders marked by vascular injury, which can result in vision impairment and blindness. The pathogenesis of many of these diseases are associated with inflammation, dysregulation of vascular permeability, angiogenesis, and ischemia, all of which are processes where ANGPTL4 is involved [64]. Diabetic retinopathy (DR) is characterized by ischemia, inflammation, and pathological neovascularization driven by chronic hyperglycemia and is one of the leading causes of blindness worldwide [110,111]. Yokouchi et al. demonstrated that high glucose induces ANGPTL4 expression in retinal pigment epithelial cells, promoting angiogenesis [112]. Perdiguero et al. showed that ANGPTL4 influences vascular maturation through endothelial markers such as caveolin-1 [113]. Clinically, ANGPTL4 levels are elevated in the aqueous and vitreous humor of patients with proliferative diabetic retinopathy, independent of vascular endothelial growth factor (VEGF), highlighting its distinct and complementary role in disease progression [72,114]. Mechanistically, hyperglycemia induces retinal hypoxia, stabilizing HIF-1α, which in turn upregulates ANGPTL4 and contributes to increased vascular permeability [115]. Lu et al. found that ANGPTL4 mediates angiogenesis downstream of the profilin-1 pathway under hypoxic and hyperglycemic conditions [116]. ANGPTL4 also binds neuropilin-1 and -2 on endothelial cells, activating RhoA/ROCK signaling and disrupting endothelial junctions [117]. Wang et al., using single-cell RNA sequencing, further revealed that ANGPTL4 modulates ligand–receptor interactions—specifically ANGPTL4–Sdc4—between vascular endothelial and photoreceptor cells, exacerbating vascular dysfunction in DR [118]. One of the clinicopathologic conditions of DR and a major cause of vision impairment is diabetic macular edema (DME), which is the thickening of the retina due to fluid retention [119]. Xu et al. found that aqueous levels of ANGPTL4 correlate with the diagnostic metrics of DME and might be able to predict the development of DME in patients with DR [120].

Beyond the diabetic context, ANGPTL4 is implicated in retinal ischemia and pathological neovascularization in retinopathy of prematurity (ROP) [121]. It is also upregulated in the aqueous humor of patients with neovascular age-related macular degeneration (nAMD), where it promotes choroidal neovascularization and may contribute to anti-VEGF resistance [122]. Chen et al. demonstrated that ANGPTL4 expression in nAMD facilitates endothelial–mesenchymal transition (EndoMT), promoting neovascular fibrosis and further compromising vascular integrity [123].

Overall, ANGPTL4 plays a critical role in mediating angiogenesis and vascular dysfunction across a spectrum of retinal diseases, particularly in diabetic and ischemic conditions [64]. Its involvement in VEGF-independent pathways highlights its promise as both a biomarker and a therapeutic target in retinal vascular disease.

### 2.3. Cancer

The functions of ANGPTL4 in lipid metabolism regulation, angiogenesis, and vascular permeability make this protein an attractive target in cancer research. Its potential roles in oncogenesis, tumorigenesis, and metastasis, as well as its potential as a diagnostic or prognostic biomarker, have been central to numerous studies across different cancer types [124]. Accordingly, ANGPTL4 has been shown to play central roles in multiple cancer types, but with widely varying functions and outcomes—ranging from anti-tumor to metastatic and oncogenic effects [125,126] (Figure 2). Therefore, although the relevance of ANGPTL4 in human cancer is undeniable and positions the protein as a possible therapeutic target, it demands caution as the functions appear to be tissue- and cancer-specific.

#### 2.3.1. Breast Cancer

In breast cancer, overexpression of ANGPTL4 correlates with tumor progression and increased malignancy, and patients with higher ANGPTL4 expression tend to have a worse overall prognosis [127]. Nevertheless, in triple-negative breast cancer (TNBC) patients, strong ANGPTL4 expression is associated with a more favorable prognosis compared to tumors with lower expression levels. In vitro, overexpression of ANGPTL4 in TNBC human cell lines reduces cell migration and invasion—a phenotype that may be mediated by the downregulation of key extracellular matrix (ECM) genes [128]. In obesity-driven breast cancer, NLRC4 inflammasome activation in tumor-infiltrating macrophages enhances ANGPTL4 expression in mammary adipocytes via IL-1β, thereby driving tumor progression and angiogenesis [13]. Moreover, Bucher et al. found that adipocyte conditioned media from adipose tissue isolated from obese mice could induce a more aggressive phenotype in TNBC cells via upregulation of ANGPTL4 and FAK [129]. Additionally, STAT3 activation in cancer-associated fibroblasts increases the secretion of ANGPTL4 into the tumor microenvironment, promoting breast cancer progression in vivo and enhancing invasion and migration in breast cancer cells [130]. Padua et al. demonstrated that transforming growth factor β (TGFβ) can induce ANGPTL4 in breast cancer cells, which promotes metastatic seeding in the lung by increasing lung vessel permeability [12]. This mechanism was corroborated by a similar finding in melanoma that tumor-secreted cANGPTL4 disrupts endothelial cell–cell junctions by directly interacting with integrin α5β1, VE-cadherin, and claudin-5, thereby promoting vascular leakiness in the tumor microenvironment—a phenomenon that facilitates lung metastasis [75]. Gong et al. later found that astrocyte-derived TGF-β2 increases ANGPTL4 expression in TNBC, promoting metastatic growth in the brain in vivo [131]. Anti-metastatic functions have also been described in melanoma mouse models, a phenotype that is mediated by increased levels of systemic nANGPTL4 derived from the primary tumor site, suggesting a correlation between serum levels of nANGPTL4 and reduced disease progression [132].

#### 2.3.2. Renal Cell Carcinoma

Renal cancer tumors exhibit the highest expression of ANGPTL4 compared to other solid human tumors. The most common subtype, clear cell renal cell carcinoma (ccRCC), shows increased ANGPTL4 expression relative to normal tissue [62]. ccRCC is characterized by loss of function of the Von Hippel-Lindau (VHL) gene, leading to the stabilization of hypoxia-inducible factors (HIF-1 and HIF-2). HIF-1 is a known transcription factor that drives ANGPTL4 expression [11,133]. However, some ccRCC tumors are HIF-2 exclusive, which points out a possible function of HIF-2 in regulating ANGPTL4 expression, which has also been shown in breast cancer models through an HIF-2-dependent upregulation of specific lncRNA [134,135]. Consequently, serum levels of ANGPTL4 may serve as a potential diagnostic biomarker for patients with RCC [136]. Nevertheless, a subset of ccRCC patients present without elevated levels of ANGPTL4 and demonstrate a poorer prognosis compared to those with elevated expression. In fact, loss of ANGPTL4 increases soft agar colony formation of ccRCC cell lines in vitro and tumor growth in vivo [62]. This tumor-suppressive phenotype is mediated by the nANGPTL4 signal peptide through a novel intracellular mechanism that inhibits LAL activity, potentially leading to modulation of lipid and/or glucose metabolism [62]. However, not all ccRCC cells lines exhibit this phenotype. While loss of ANGPTL4 in 786O ccRCC cells increased soft agar colony formation in vitro and increased lysosomal acid lipase activity, it decreased in vivo tumor growth, as did treatment of mice with an antibody targeting cANGPTL4 [62]. This indicates that the role of ANGPTL4 in ccRCC is complex and nANGPTL4 may act as a tumor suppressor in a subset of samples, while cANGPTL4 can promote tumor progression. More studies of ANGPTL4 in ccRCC need to be performed to further elucidate its function and potential as a therapeutic target.

#### 2.3.3. Prostate Cancer

Although patients diagnosed with localized prostate cancer have a 10-year life expectancy of nearly 99%, one of the main challenges lies in accurately identifying patients with clinically significant disease. This is essential to avoid treatment-related comorbidities, unnecessary surgeries, and increased healthcare costs [137,138]. In this regard, commonly used biomarkers such as serum levels of prostate-specific antigen (PSA) lack sufficient predictive value [138,139]. In prostate cancer, tumor hypoxia has been shown to promote aggressiveness and induce androgen-independent growth of prostate cancer cells in vitro [140]. Chronic hypoxia increases the expression of ANGPTL4 in LNCaP cells, and its overexpression promotes cell migration, activation of PI3K/AKT survival pathways, and resistance to apoptosis and drug treatment via Bcl-2 upregulation [140]. Moreover, normal prostate tissue does not express ANGPTL4, and its positive expression—when combined with tumor stage—serves as an indicator of PSA recurrence following surgery [140]. Therefore, ANGPTL4 represents a promising biomarker and a potential therapeutic target in the diagnosis and treatment of prostate cancer. Within the tumor microenvironment, cancer-associated fibroblasts (CAFs) secrete ANGPTL4, which binds to IQGAP1 to activate RAS–ERK signaling. This ultimately induces the expression of PGC1α, which enhances mitochondrial functions such as oxidative phosphorylation—a mechanism that may contribute to prostate cancer chemoresistance [141].

#### 2.3.4. Pancreatic Cancer

Chronic inflammation associated with pancreatitis is a well-established risk factor for the development of pancreatic ductal adenocarcinoma (PDAC) [142]. In this context, ANGPTL4 has been shown to induce pancreatitis in mouse models by upregulating complement component 5a (C5a) in macrophages via PI3K/AKT signaling [143]. Treatment of mice with severe and mild pancreatitis using an ANGPTL4-neutralizing antibody results in reduced C5a levels and significantly alleviates pathological features of pancreatitis [143]. Another study demonstrated that ANGPTL4 expression is elevated in acinar-to-ductal metaplasia (ADM) lesions and pancreatic intraepithelial neoplasia (PanIN), both of which are recognized as precursors of PDAC. Furthermore, overexpression of ANGPTL4 in KRAS-mutant pancreatic cancer cells enhances carcinogenesis, cell proliferation, and the formation of metastatic lesions. This oncogenic phenotype is mediated by ANGPTL4-dependent upregulation of the extracellular matrix protein periostin [144].

#### 2.3.5. Lung Cancer

In non-small cell lung cancer (NSCLC), ANGPTL4 is significantly overexpressed compared to normal tissue. ANGPTL4 has been shown to enhance proliferation, migration, and invasion of NSCLC cells through ERK1/2 signaling and regulation of epithelial–mesenchymal transition (EMT) [145]. Additionally, ANGPTL4 regulates cellular metabolism in NSCLC by promoting glutamine and fatty acid oxidation, thereby fueling oxidative phosphorylation [146]. ANGPTL4 has been shown to promote lung adenocarcinoma (LUAD) cell proliferation, migration invasion, and lipid production; conversely, knockdown of ANGPTL4 inhibited migration and apoptosis [147,148]. In LUAD cells, ANGPTL4 expression has also been linked to acquired resistance to the EGFR-TKI gefitinib, a mechanism mediated by the regulation of pyroptosis and apoptosis—suggesting its potential as a therapeutic target to prevent tyrosine kinase inhibitor (TKI) resistance [149].

#### 2.3.6. Colorectal Cancer

Previous studies have shown that ANGPTL4 expression is weak in normal colorectal tissue; however, more than 50% of human colorectal adenocarcinoma samples display positive staining for ANGPTL4 [150]. Although ANGPTL4 expression does not correlate with overall survival, it is significantly associated with tumor invasion, indicating a potential role in metastasis. More recently, colorectal cancer (CRC) cells have been shown to activate hepatic stellate cells (HSCs) via FGF-19, promoting their differentiation into cancer-associated fibroblasts (CAFs). This, in turn, increases ANGPTL4 expression and secretion, enhancing CRC cell migration and promoting liver metastasis [151]. Additionally, adipose-derived stem cells (ADSCs) stimulate ANGPTL4 expression in CRC cells via TGF-β signaling through the transcription factor SMAD3. This activation leads to increased glycolysis, enhanced migration, and resistance to anoikis—a phenotype that supports peritoneal metastasis of CRC [152]. Furthermore, tumors with high ANGPTL4 expression are associated with higher rates of KRAS mutation, increased PI3K–AKT signaling, and elevated expression of glucose transporters GLUT1 and GLUT3, suggesting a role for ANGPTL4 in regulating glucose metabolism and contributing to CRC progression [153].

## 3. Discussion

In the 25 years since first being discovered [1,3], much has been learned about the biochemical and biological function of ANGPTL4 and its role in the pathogenesis of several human diseases. Despite this robust field of knowledge, much still needs to be discovered. Early studies discovered the role of ANGPTL4 in regulating lipid metabolism and the mechanism of how it regulates the availability of fatty acids and cholesterol systemically during fasting and feeding states has been well characterized (Figure 1B). However, recent studies have indicated that ANGPTL4 inhibits the activity of other lipases such as endothelial lipase [61], lysosomal acid lipase [62], and hepatic lipase [154], indicating that ANGPTL4 may be involved in regulating fatty acid bioavailability at a tissue or cell specific level. The mechanism(s) of how ANGPTL4 regulates these other lipases, either directly such as it does with LPL or through some indirect mechanism, needs to be further studied. Many other functions of ANGPTL4 have been described as well, such as regulating the survival and/or differentiation of stem/progenitor cells, regulating inflammation, promoting endothelial vessel permeability, and regulating angiogenesis (Figure 1C). Both pro- and anti-angiogenic functions for ANGPTL4 have been reported [63,64]. Some of these contradicting reports can be due to the role of cleaved cANGPTL4 versus full-length ANGPTL4, as several studies have shown that full-length ANGPTL4 can interact with proteoglycans in the extracellular matrix and inhibit angiogenesis [65,66]. However, the mechanism(s) and conditions in which ANGPTL4 regulates angiogenesis require further studies. Recent evidence indicates that ANGPTL4 may generate a more metabolically favorable phenotype for angiogenesis by regulating endothelial cell metabolism through inhibition of endothelial lipase [61]. Thus, the function of nANGPTL4 may facilitate the pro-angiogenic function of cANGPTL4.

ANGPTL4 has been shown to play a role in the pathophysiology of several human diseases including atherosclerosis, various retinopathies, and several cancers. The role ANGPTL4 plays in these conditions is variable and can either be protective or deleterious. In atherosclerosis and ischemia, ANGPTL4 has been shown to play a protective role by promoting increased blood flow to the sites, stabilizing atherosclerotic plaques, and inhibiting inflammation [78,99,103,155]. Thus, ANGPTL4 has been proposed as a biomarker and a potential therapeutic. However, this is complicated by studies showing that loss of function of ANGPTL4 or treatment with a blocking antibody significantly lowers hypertriglyceridemia, a known risk factor for atherosclerosis [32,33,93,156], and studies have correlated ANGPTL4 levels with the risk of heart failure [108]. ANGPTL4 plays a key role in the pathogenesis of various retinopathies, including diabetic retinopathy, and several studies have suggested that targeting ANGPTL4, in particular cANGPTL4, may be beneficial in combination with VEGF-targeted therapies [72,116,122,157]. In cancer, ANGPTL4 has been shown to play varied roles depending on the type of cancer. It can promote angiogenesis and disease progression in obesity-driven breast cancer and ccRCC, promote metastasis in the lung and brain, promote survival and circulating cancer cells, and regulate tumor metabolism (Figure 2). The regulation of tumor metabolism can potentially both promote tumor progression, as in the case of colorectal cancer [153], or suppress tumor progression, as shown in ccRCC [62]. Thus, ANGPTL4 may be a potential therapeutic target in cancer, but the context in which it may be beneficial needs further study.

Several methods to target ANGPTL4 have been deployed. Monoclonal antibodies against ANGPTL4 (REGN1001) have been shown to reduce triglyceride levels in both mice and non-human primates [33,158]. These antibodies were generated by immunizing ANGPTL4 knockout mice with full length ANGPTL4. Other groups have developed antibodies specifically against cANGPTL4. The Tan laboratory generated antibodies against cANGPTL4 by immunizing rabbits with recombinant cANGPTL4 [159]. They have since shown that these antibodies can impair tumor growth in vivo and induce apoptosis of cancer cells in non-adherent conditions [160]. Furthermore, they have reported that antibodies against cANGPTL4 can reduce pulmonary edema in secondary pneumonia [161] and increase lung tissue recovery in influenza pneumonia [85]. Our laboratory has also developed monoclonal antibodies against cANGPTL4 that can inhibit cANGPTL4-induced angiogenesis both in vivo and in vitro [13]. We have further reported that one of these antibodies, clone 6A11A7, can reduce 786O tumor growth and angiogenesis in mice xenograft models [13,62]. The further development of antibodies targeting ANGPTL4 is a promising area of research for the treatment of several diseases, including cancer, pneumonia, and retinopathies.

## 4. Limitations

This review has certain limitations. It primarily focused on the roles of ANGPTL4 in cancer, while other important areas, such as cardiovascular and metabolic diseases, received comparatively less coverage. Two recent studies have suggested that ANGPTL4 may play a role in hemangiomas, which are benign growths of extra blood vessels. Hemangiomas can be present at birth (infantile) and often appear as “strawberry marks” or birthmarks [162]. Abnormal outgrowth of blood vessels can also occur due to viral infections such as SARS-CoV-2 [163,164,165]. The beta-blocker propranolol is the most common drug used to treat hemangiomas in infants, though the mechanism of action is not understood [166]. Recently it was shown that propranolol can be used to suppress the formation of hemangiomas in the lungs of COVID-19 patients by suppressing the expression of ANGPTL4 [165], providing a potential mechanism of action for propranolol in treating hemangiomas. Broader discussion of the role of ANGPTL4 in hemangiomas and other pathologies with a limited number of studies on the role of ANGPL4 have largely been left out of this review. Furthermore, much of the available evidence comes from preclinical or observational studies, and translation into clinical applications is still limited. Future research should aim to resolve these uncertainties, expand the understanding of ANGPTL4’s diverse functions, and explore its potential as a therapeutic target across a broader range of human diseases.

## 5. Conclusions

Over the past 25 years, significant efforts have been made in understanding the diverse roles of ANGPTL4 in lipid metabolism, angiogenesis, vascular permeability, inflammation, and cancer biology. ANGPTL4 has emerged as a potential biomarker and therapeutic target in multiple diseases, although its functions remain complex and context-dependent, with multiple effects reported across different tissues and conditions.

## Figures and Tables

**Figure 1 cancers-17-02364-f001:**
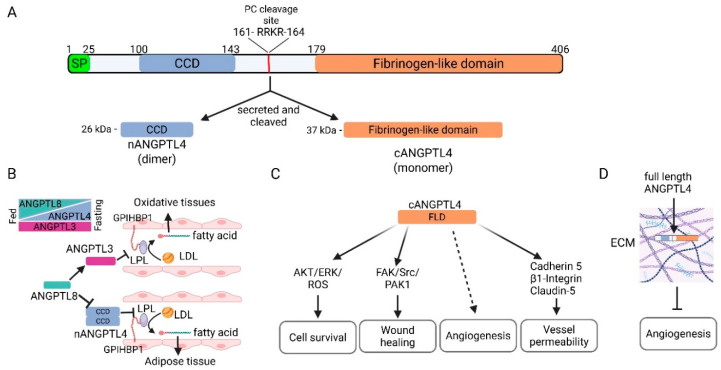
ANGPTL4 structure and function. (**A**) ANGPTL4 is a secreted glycoprotein that consists of an N-terminal coiled-coil domain (CCD), a protein convertase (PC) cleavage site, and a C-terminal fibrinogen-like domain (FLD). Upon secretion, it is cleaved by PCs to form nANGPTL4 (as a dimer) and cANGPTL4. (**B**) nANGPTL4 in coordination with ANGPTL3 and 8 is involved in the systemic regulation of lipid metabolism during feasting and fasting states to control the availability of free fatty acids and cholesterol. Under a fasting state, ANGPTL4 levels increase in adipose tissue and ANGPTL8 levels decrease, leading to increased inhibition of lipoprotein lipase (LPL) by nANGPTL4, decreased fatty acid uptake by adipocytes, and increased circulating lipoproteins. Concordantly, decreased circulating ANGPTL8 undermines the ability of circulating ANGPTL3 to inhibit LPL in oxidative tissues, leading to increased lipoprotein metabolism and increased fatty acid uptake. The opposite occurs in a fed state. (**C**) Multiple functions of cANGPTL4 have been reported, including promotion of cell survival, promotion of wound healing, angiogenesis, and regulation of vessel permeability. (**D**) Full length ANGPTL4 that is not cleaved when secreted can bind to proteoglycans in the extracellular matrix (ECM) and inhibit angiogenesis. Created in BioRender. Kolb, R. (2025) https://BioRender.com/t66a408.

**Figure 2 cancers-17-02364-f002:**
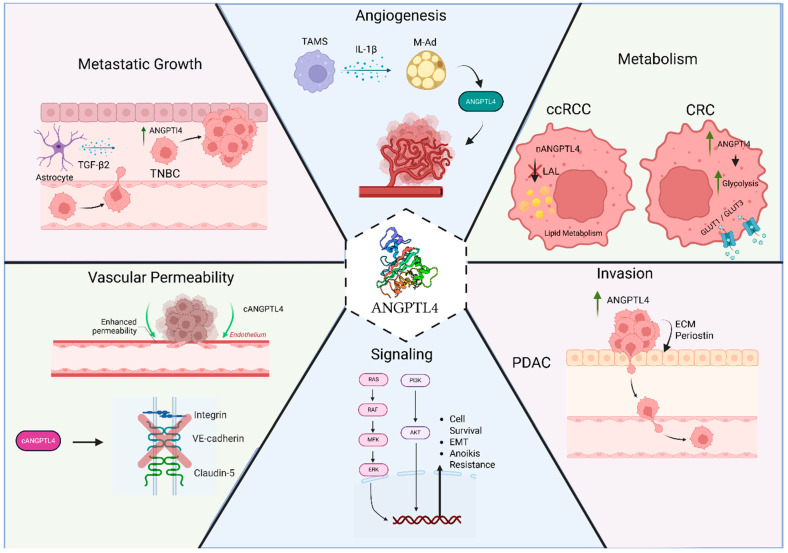
Function of ANGPTL4 in cancer. The function of ANGPTL4 in cancer varies depending on the type of cancer. It has been reported to promote angiogenesis in obesity-driven breast cancer progression and clear cell renal cell carcinoma (ccRCC), regulate cancer cell metabolism in ccRCC and colorectal cancer (CRC), promote invasion of pancreatic cancer, promote cell survival and epithelial–mesenchymal transition (EMT) in circulating cancer cells, promote seeding of lung metastasis by increased lung vessel permeability, and promote brain metastasis. Created in BioRender. Ramos, P. (2025) https://BioRender.com/op3wn94.

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
