# Peer review of "ANGPTL4: A Comprehensive Review of 25 Years of Research"

_cancers, 2025, doi:10.3390/cancers17142364_

Round 1
Reviewer 1 Report
Comments and Suggestions for Authors
A review of the function of Angptl4 is timely, but the manucsript could be improved by the following
1) In the discussion of ccRCC, HIF2 should be mentioned, and Angplt4 may be more of a HIF2 target gene than HIF1. There are ccRCC which completely lack expression of HIF1, but there is no ccRCC t hat lacks expression of HIF2. This should be corrected. Also, the situation of decreased soft agar growth but increases in vivo growth has been described before, pls add reference ( Inhibition of MAP kinase kinase causes morphological reversion and dissociation between soft agar growth and in vivo tumorigenesis in angiosarcoma cells.
.Am J Pathol. 2000 Dec;157(6):1937-45. doi: 10.1016/s0002-) 2) Mention of the role of Angptl4 in hemangiomas of infancy and COVID should be made, as Angplt4 is elevated in the blood of patients with severe COVID and propranolol, the major treatment of hemangiomas, works on hemangiomsa to reduce Angptl4 by a beta blocker independent mehcanism (Propranolol exhibits activity against hemangiomas independent of beta blockade. .NPJ Precis Oncol. 2019 Nov 1;3:27. doi: 10.1038/s41698-019-0099-9. eCollection 2019.PMID: 31701018, R-Propranolol Has Broad-Spectrum Anti-Coronavirus Activity and Suppresses Factors Involved in Pathogenic Angiogenesis. . Int J Mol Sci. 2023 Feb 27;24(5):4588. doi: 10.3390/ijms24054588.PMID: 36902020 Free PMC article. 3) An anti-metastatic role fo the n-terminus of ANgptl4 has been found (Primary tumor-derived systemic nANGPTL4 inhibits metastasis. .J Exp Med. 2023 Jan 2;220(1):e20202595. doi: 10.1084/jem.20202595. Epub 2022 Oct 21.PMID: 36269299 Free PMC article.)
Author Response
Thank you very much for taking the time to review this manuscript. Please find the detailed responses below and the corresponding revisions/corrections highlighted in the re-submitted files
Comments 1: In the discussion of ccRCC, HIF2 should be mentioned, and Angplt4 may be more of a HIF2 target gene than HIF1. There are ccRCC which completely lack expression of HIF1, but there is no ccRCC that lacks expression of HIF2. This should be corrected. Also, the situation of decreased soft agar growth but increases in vivo growth has been described before, pls add reference ( Inhibition of MAP kinase kinase causes morphological reversion and dissociation between soft agar growth and in vivo tumorigenesis in angiosarcoma cells.
Response 1: Thank you for making this point. We agree with this comment. Therefore we have included a short mention about HIF2 in page 10 under the Renal Cell carcinoma section. This includes a statement pointing that no ccRCC lacks HIF2 expression and a mention to a previous study that points out how HIF2 is involved in ANGPTL4 expression through HIF2 dependent upregulation of a specific lncRNA. As for the phenotype that is mentioned, in this case the observed phenotype is increased soft agar growth in vitro and increased tumor growth in vivo while in a different cell line is the same phenotype but reduced tumor growth in vivo in both cases colony formation in vitro is increased.
Comment 2: Mention of the role of Angptl4 in hemangiomas of infancy and COVID should be made, as Angplt4 is elevated in the blood of patients with severe COVID and propranolol, the major treatment of hemangiomas, works on hemangiomsa to reduce Angptl4 by a beta blocker independent mehcanism (Propranolol exhibits activity against hemangiomas independent of beta blockade. .NPJ Precis Oncol. 2019 Nov 1;3:27. doi: 10.1038/s41698-019-0099-9. eCollection 2019.PMID: 31701018, R-Propranolol Has Broad-Spectrum Anti-Coronavirus Activity and Suppresses Factors Involved in Pathogenic Angiogenesis. . Int J Mol Sci. 2023 Feb 27;24(5):4588. doi: 10.3390/ijms24054588.PMID: 36902020 Free PMC article.
Response 2: This is a good suggestion from the reviewer. A very brief discussion of this was added to the revised manuscript in the limitations sections. Due to the limited number of studies on this, it was not discussed in more detail and added as a limitation of the review.
Comments 3: An anti-metastatic role fo the n-terminus of ANgptl4 has been found (Primary tumor-derived systemic nANGPTL4 inhibits metastasis. .J Exp Med. 2023 Jan 2;220(1):e20202595. doi: 10.1084/jem.20202595. Epub 2022 Oct 21.PMID: 36269299 Free PMC article.)
Response 3: We agree with this and we have now incorporated this reference in page 10 under the breast cancer section where we focus on metastasis and we briefly introduce some findings on melanoma models as well which is what this reference focuses on.
Reviewer 2 Report
Comments and Suggestions for Authors
The manuscript well-summarizes the biological functions of ANGPTL4, including regulation of lipid metabolism, angiogenesis, and vascular permeability, and other functions such as inflammation, cell survival, and differentiation. The authors also discussed the role of ANGPTL4 in various diseases, including cardiovascular diseases, retinopathies, and cancers.
- Please briefly introduce the ANGPTL family and the difference between ANGPTL4 and other ANGPTL members.
- The FLD domain is conserved across the ANGPTL family, and the FLD domain mediates ANGPTL4’s roles in cell survival, wound healing, and angiogenesis. Is ANGPTL4’s role in angiogenesis and vessel permeability unique to ANGPTL4 or not? What is the difference between cANGPTL4 and the FLD domain (the cleaved form) of other ANGPTLs?
- Several studies reported a positive correlation between tumor progression and ANGPTL4 levels in many cancers. Which form of ANGPTL4 is tested, cANGPTL4 or nANGPTL4?
- ANGPTL4 expression can be regulated by various factors, including PPARs, transcription factors, and cytokines. Is the cleavage of ANGPTL4 regulated, e.g., via the expression of pro-protein convertases?
Author Response
Thank you very much for taking the time to review this manuscript. Please find the detailed responses below and the corresponding revisions/corrections highlighted in the re-submitted files
Comments 1: Please briefly introduce the ANGPTL family and the difference between ANGPTL4 and other ANGPTL members
Response 1: This is an excellent suggestion from the reviewer and a brief discussion of teh ANGPTL family of proteins and where ANGPTL4 fits in has been added to section 1 - structure and function - of the revised manuscript.
Comments 2: The FLD domain is conserved across the ANGPTL family, and the FLD domain mediates ANGPTL4’s roles in cell survival, wound healing, and angiogenesis. Is ANGPTL4’s role in angiogenesis and vessel permeability unique to ANGPTL4 or not? What is the difference between cANGPTL4 and the FLD domain (the cleaved form) of other ANGPTLs?
Response 2: The unique and shared functions of the FLD domain of cANGPTL4 was added in section "1.2. Angiogensis and vascular permeability"
Comments 3: Several studies reported a positive correlation between tumor progression and ANGPTL4 levels in many cancers. Which form of ANGPTL4 is tested, cANGPTL4 or nANGPTL4?
Response 3: Most of these studies which look at ANGPTL4 expression across tissue samples, cancer cell lines, patient data look at intracellular levels or full length ANGPTL4 (mRNA, protein levels). Once the protein is secreted then is subjected to processing through proprotein convertases into nANGPTL4 and cANGPTL4 distinctively. Therefore, some studies look directly at the intracellular effects that ANGPTL4 has (metabolism, colony growth, etc.) while some studies look at the effect that each specific c-terminal and n-terminal domain has within the tumor microenvironment once the protein is secreted (metastasis, CAFs, macrophages).
Comments 4: ANGPTL4 expression can be regulated by various factors, including PPARs, transcription factors, and cytokines. Is the cleavage of ANGPTL4 regulated, e.g., via the expression of pro-protein convertases?
Response 4: Thank you for making this point. We completely agree. In the manuscript we point out how the processing of ANGPTL4 once is secreted could relate to its tissue-specific function. Since the processing is regulated by pro-protein convertases the expression of these could have an impact. Therefore, we have now included in page 2 under the structure and function section a small comment on some of the PCs that process ANGPTL4 and how the expression of these could be involve in regulating ANGPTL4 specific functions.
Reviewer 3 Report
Comments and Suggestions for Authors
The peer-reviewed manuscript "ANGPTL4: A Comprehensive Review of 25 Years of Research" is a comprehensive review of ANGPTL4. It is very well written, with good selection of references. However, the publication focuses on results from cancer.
I have some minor comments:
- I miss the review methodology and the rules for selecting references
- the work lacks information on the effect of ANGPTL4 on metabolic diseases
- it would be worth adding in the summary/discussion some limitations of the review, such as the fact that the described results mainly concern cancer diseases
Author Response
Thank you very much for taking the time to review this manuscript. Please find the detailed responses below and the corresponding revisions/corrections highlighted in the re-submitted files.
Comments 1: I miss the review methodology and the rules for selecting references.
Response 1: Thank you for this valuable comment. We acknowledge that a specific section pointing out methodology and reference selection was not made in our manuscript. We constructed this manuscript as a narrative review rather than a systematic review. As so we selected references based on keyword searches through PubMed and Google Scholar giving priority to peer-reviewed articles published in the last 10 years as well as those with historical importance in the topic. We agree it would be helpful to clarify this in the manuscript. We have now added a statement as "Literature Search" right after the Conclusions explaining that this is a narrative review and that references were selected based on relevance and recency to cover the broad scope of ANGPTL4 research.
Comments 2: the work lacks information on the effect of ANGPTL4 on metabolic diseases
Response 2: Thank you for pointing this out. We agree that our manuscript lacks a dedicated section expanding on the effect of ANGPTL4 on metabolic diseases. However, our primary focus in this review was on the role of ANGPTL4 in cancer. While we do discuss certain effects of ANGPTL4 on metabolism, these are mainly presented in the context of cancer biology, tumor progression and structure/function of the protein. Other topics, such as ANGPTL4’s roles in vascular permeability, angiogenesis, and its involvement in other human diseases, were included because of the relevance of the research and the context/field in which research on ANGPTL4 is mainly conducted. We acknowledge that a more detailed discussion of ANGPTL4 in purely metabolic contexts was beyond the intended scope of this manuscript. Nevertheless, we have now added this specific point as a limitation in the conclusions sections to clarify the focus of our review.
Comments 3: it would be worth adding in the summary/discussion some limitations of the review, such as the fact that the described results mainly concern cancer diseases.
Response 3: We agree that it is important to acknowledge the limitations of the review including the fact that the majority of the described results relate to cancer. In response, we have included these in the conclusion section to explicitly state this limitation and clarify that while ANGPTL4 is discussed in the context of various biological processes and diseases, the main focus of this review is on its roles in cancer.